

# Behavior change intervention to sustain iodide salt utilization in households in Ethiopia and study of the effect of iodine status on the growth of young children: community trial

Abebe Ferede[1], Muluemebet Abera Wordofa[2] and Tefera Belachew[2]

[1] Department of Public Health, Arsi University, Asella, Ethiopia
[2] Department of Population and Family Health, Jimma University, Jimma, Ethiopia

Corresponding author
Abebe Ferede, abebeferede95@yahoo.com

## ABSTRACT

**Background**. Monitoring systems in a broad range of countries are a notable effort to eliminate iodine deficiency disorders (IDDs). This study aimed to gather data on the amount of iodide present in table salt and how household consumption patterns affect children's iodine status and its effect on their growth.

**Methods**. A single treatment arm community trial study design was designed. Lower community units (LCUs) were chosen at random from districts assigned either intervention or control. From a list of LCUs, 834 mothers and their paired children were chosen randomly. Urine and table salt samples were collected and examined in the national food and nutrition laboratory. The deference between arms was determined using a $t$ test, and the generalized estimating equation (GEE) was used to forecast parameters.

**Results**. The mean iodide content in the table salt samples of 164 (98.1%) was 45.3 ppm and a standard deviation (SD) of 14.87, which were above or equal to the recommended parts per million (ppm). Between the baseline survey and the end-line survey, the mean urine iodine concentration (UIC) was 107.7 µg/L ($+/- 8.64$ SD) and 260.9 µg/L ($+/- 149$ SD). Children's urine iodine excretion (UIE) had inadequate iodine in 127 (15.2%) children at the beginning of the study, but only 11 (2.6%) of the intervention group still had inadequate iodine at the end. The childrens' mean height (Ht) was 83.1 cm ($+/-10$ SD) at baseline and 136.4 cm ($+/-14$ SD) at the end of the survey. Mothers knew a lot (72%) about adding iodized salt to food at the end of cooking, and 183 (21.9%) of them did so regularly and purposefully. A total of 40.5% of children in the intervention group had stunted growth at baseline, which decreased to 15.1% at the end of the study but increased in the control group to 51.1%. The mean difference (MD) of urine iodine concentration (UIC) between intervention and control groups was 97.56 µg/L, with a standard error (SE) of 9.83 ($p = 0.001$). The end-line Ht of children in the intervention group was increased by 7.93 cm ($\beta = 7.93$, $p = 0.005$) compared to the control group.

**Conclusions**. Our research has shown that mothers who embraced healthy eating habits had perceived improvements in both the iodine status and height growth of their children. In addition to managing and using iodine salt, it has also introduced options for other healthy eating habits that will also play a significant role in their children's future development. This sort of knowledge transfer intervention is essential for the
sustainability of society's health. Therefore, this trial's implications revealed that the intervention group's iodine status and growth could essentially be improved while the control group continued to experience negative effects.
**Trial registration**. ClinicalTrials.gov Identifier: NCT048460 1.

## INTRODUCTION

Iodine deficiency is the most prevalent nutritional issue owing to its urgency, severity, and excessive impact on public health (*Weerasekara et al., 2020*). The urgency to treat iodine deficiency disorders (IDD) has enhanced universal iodinated salt intervention programs. But not concerning the steadiness of the quantity of iodine found in iodinated salt in the households of the distinct geographical areas (*Iodine Global Network, 2017*). The requirement for iodine dietary allowance for young children is very high compared to the estimated energy requirement (EER) for adults. Even among young children, those aged less than 12 months need more iodine than those aged 12–36 months, and these children need more iodine than children aged 37–59 months (*Bath et al., 2022*). The iodine requirement in children starts with early brain development, especially at the early age of two years, when brain development begins in early pregnancy and continues for the first years of life (*Prado & Dewey, 2014*). It is also crucial to start a dietary iodine intake change due to behavioral communication; the brain actually recovers as soon as possible to prevent the compromised growth process from happening when young children experience brain damage due to early iodine deficiency (*Prado & Dewey, 2014*). Early iodine deficiency can cause brain damage, but it can be reversed with behavioral modification that improves women's attitudes toward using iodized salt and encourages households (HH) to use it at home by offering better options for food preparation (*GAIN—UNICEF, 2018*). The most crucial group for the appropriate use of iodinated salt in HH and the long-term eradication of IDD are women (*GAIN—UNICEF, 2018*). The varying degrees of iodine loss in table salts reached up to 30% between production and consumption (*WHO, 2014*). In addition, a study showed that iodine levels in all refined salt samples decreased under various environmental conditions but not in dark or non-humid environments (*Fallah et al., 2020*). Furthermore, the estimated iodide status of young children and the quantified iodine in table salt in the household showed a wide gap at baseline. Therefore, programmatic monitoring of iodide salt intervention at the household level is essential to prevent public health risks from iodine deficiency (*Iodine Global Network, 2017*; *UNICEF, 2021*). In addition, a nutrition behavior intervention approach is important for reminding households about their sensitivity to iodinated salt and its persuasion. To fill these gaps, evaluating and enhancing women's knowledge and attitudes regarding IDD prevention and control in the community is a fundamental building block for places where communication and literacy barriers are prevalent (*The United Nations Educational, Scientific and Cultural Organization, 2017*).

Iodine deficiencies in Ethiopia have been consistently high in recent decades because of poor knowledge and attitudes toward utilizing iodized salt in households, insufficient food availability, poor maternal and child nutrition, and sociopolitical and economic factors (*Bonglaisin et al., 2019*). Therefore, measuring UIC frequently in young children to protect them from subclinical iodine deficiency and assess their current iodine status and the effects of iodine concentration on their growth is beneficial to the nation.

### The general objective

The general objective of this study was to sustain household iodide salt utilization and iodine status in young children and study the effect on their growth.

  Specific objectives of the study.

1. Evaluate the iodide content of the iodized table salt sold and used in households.
2. Assess the knowledge of mothers about the utilization of iodized salt.
3. Determine the baseline iodine content of excreted urine in children aged 6–59 months.
4. Execute behavior-change interventions to sustain the use of iodide salt in households.
5. To examine the impact of iodine concentration on children's linear growth at the end line.
6. Evaluation of children's slower growth in the intervention and control groups with the iodine content of their expelled urine.

## MATERIALS AND METHODS

### Study design

A single treatment arm, community-based cluster randomized trial design was designed.

### Study period and study area

This study started on April 25, 2021, and ended on October 30, 2022. The study area included districts found between the Arsi and Bale zones of the Oromiya region. Specifically, Shirka, Ticho, Munessa, Hinkollo, Meraro, Hikollo Wabe, and Mechtu were districts included in the current study (Fig. 1).

### Sample size

The sample size for this study was calculated using the formula of statistical superiority design to hypothesize the statistical superiority of one group over the other following an accepted intervention (*Wang et al., 2017*). It was determined using Gpower software (version 3.0) with a power of 95%, precision of 5%, and effect size of 0.25, yielding 834.

  To evaluate the iodide status of table salt found in households (HHs), approximately 884 table salt samples were collected, of which 50 were collected from markets. A total of 884 samples were offered to the Ethiopian Food and Nutrition Laboratory, and laboratory experts designed approximately 19 (2%) samples randomly selected for pre-test analysis, 2% (17 of 834) samples collected from households, and 4% (two of 50) samples from markets, and the results were interpreted. Time spent on one sample and the cost in terms of the amount of money and quality of findings (likely to even) were fundamentally used for planning the entire work. Thus, they applied random sampling to make it more

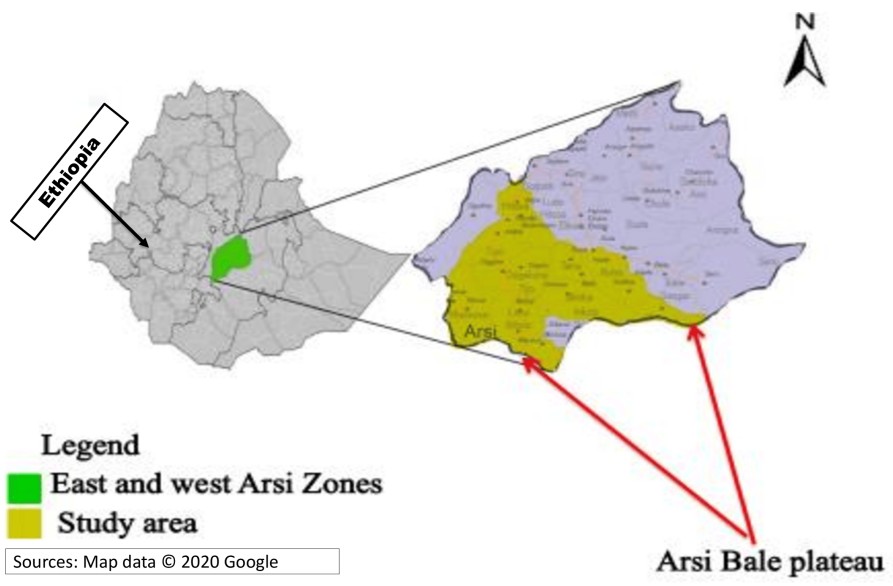

Legend

■ East and west Arsi Zones
■ Study area

Sources: Map data © 2020 Google

Arsi Bale plateau

**Figure 1 Study geographical area at national, regional and local study area level.** Sources: Map data ©2020 Google.

advantageous than total sample analysis. Therefore, $120 + 25 + 19 = 164$ was the total analyzed table salt for iodine. As a result, each low-community unit (LCU) contributed more than nine table salt samples for analysis.

### Randomization

Randomization had taken place within 16 low-community units (LCUs), which were randomly assigned to eight control clusters and eight intervention clusters. Sequence generation districts were selected based on the random assignment of LCUs. Two LCUs were randomly selected from each district, and 16 LCUs were selected from 202 LCUs to meet the study sample size. Emergency Nutritional Assessment (ENA) for SMART 2020 software was used for randomizing sixteen LCUs, allotted equally to control and intervention (See Fig. 2).

### Cluster masking

In community-based trials, the cluster masking method was required to maintain data information on the intervention cluster. Cluster masking was derived from non-selected LCUs, which served as buffer zones (*Ahmad, Abdullah & Jaafar, 2013*). The selection of a few K values from a large number of LCUs provided sufficient protection of the behavior change information from contamination with the control. That "K" is sample size, *i.e.,* selected from Kth = N/sample size = 202 LCU, 16 LCU, and 12 LCU found in each intervention or control cluster. Thus, every 12th LCU was recruited as a study cluster unit, and the study populations were obtained from these LCUs. Therefore, at least 12 LCUs were used as a boundary (mask) for each LCU included in this study.

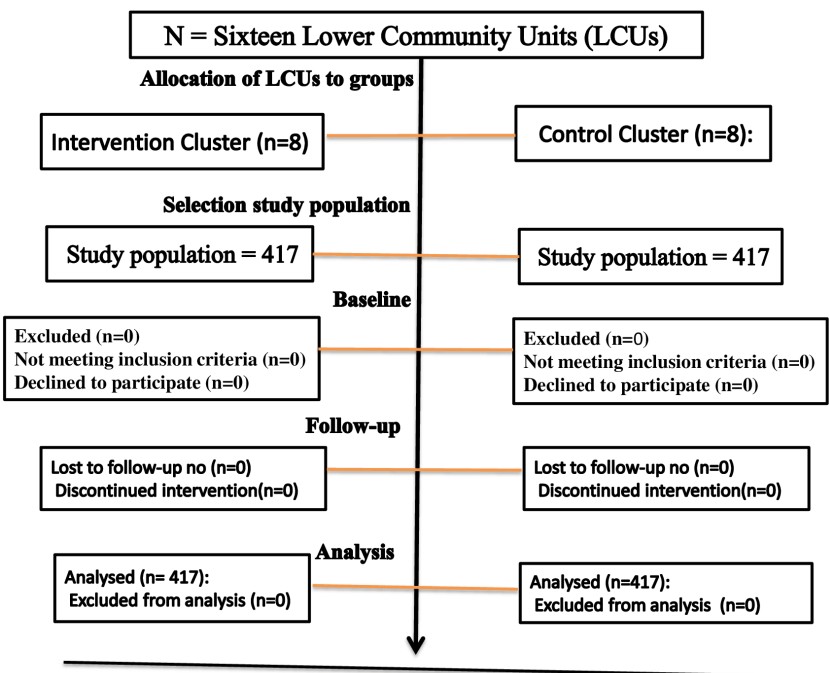

**Figure 2** The flow diagram the progress through the phase of a perioral cluster randomized trial groups' basic eligibility and allocation to the group, selection of study population, follow-up.

## Eligibility

The study participants were all children aged 6–59 months, with a pair of mothers or caregivers selected for intervention and control clusters using a systematic random sampling method from randomly assigned LCUs.

Arm 1: a control group that did not receive an iodide dietary intake behavior intervention. Control cluster: eight randomly assigned LCUs. The control group included 417 mothers or caregivers and paired children aged 6–59 months who were selected from the control cluster.

Arm 2: Intervention group that was provided iodide dietary intake behavior interventions. The intervention cluster included eight randomly assigned LCUs. The intervention group included 417 mothers or caregivers and paired children aged 6–59 months who were selected from the intervention cluster.

## Exclusion criteria

All children aged 6–59 months in the central highlands in the control group had iodine deficiency. They also had physical disabilities, including deformities and moderate-to-severe acute or chronic illnesses. Children from other areas had different characteristics and lived far away from the study area.

## Implementation of interventions

The iodide dietary intake behavior intervention was administered to the intervention groups for 17 months, with a monthly frequency for each LCU, by trained health extension

workers (HEWs). Intervention was provided to the group within each cluster. HEWs mainly implemented intervention programs aimed at strengthening the knowledge and attitudes of mothers or caregivers to improve the exploitation of table salt with iodine, reduce the prevalence of iodine deficiency, and improve the linear growth of children. Mothers or other primary caregivers participated in behavior contracts for the sustainable use of iodized salt, self-monitoring, and goal-setting for the consumption of salt with sufficient iodine. HEW's communication initiatives play a significant role. Each HEW activity received feedback, and supervision was regularly conducted. Barriers to the use of salt with iodine, such as cost, women's behavior, adverse events, suspicion of an unpleasant taste, and dietary beliefs, were addressed during the implementation of the intervention. Assigned supervisors constantly provided support and feedback on each phase of the intervention events.

## Measurements

### Procedure used to measure iodide in table salt

Each sample of table salt weighed approximately 10 g when prepared (*Zimmermann & Andersson, 2021*; *World Health Organization, 2007*). Indeed, all of the table salt samples were gathered from the present study area's geographic location. Essentially, we sent $834 + 50 = 884$ samples to the Ethiopian Food and Nutrition Laboratory (whose dataset was already submitted) for analysis. Laboratory experts randomly selected 19 (2%) samples (2% of the 17 samples collected from 834 households and 4% of the 50 samples collected from the market) for pre-test analysis. The findings were then interpreted. The time commitment for a single sample, the expenses related to the outcomes, and the quality (even) of the findings were the main drivers of their entire work. They used random sampling as a result to make it better than the challenge of total sample analysis. Therefore, in the first phase, 19 samples were analyzed, then 120 samples from households and 25 samples from the market were analyzed for general inferences. The analytical procedure exploited all the required principles and recommendations of the British Pharmacopeia to quantify iodine in table salts (*Dary, 2011*; *Machado et al., 2017*).

This process was carried out in compliance with the World Health Organization's 2007 recommendation for monitoring and titrating salt to determine its iodine content (*World Health Organization, 2007*). Based on the amount of iodide contained in ppm, the interpretation of the quantification of iodine found in table salts is primarily divided into three factions. The results were then expressed as under the standard limit (<15 ppm), between the standard allowable limits (15 and 40 ppm), and exceeding the standard limit (<40 ppm) (*Zimmermann & Andersson, 2021*; *World Health Organization, 2007*).

### Excreted urine iodine concentration determination

All 834 children provided urine samples during baseline and end-line surveys. According to WHO guidelines, the urine samples were taken, transported to the EPHI, and subjected to ammonium sulfate analysis to determine the amount of iodine excreted in the urine (*World Health Organization, 2007*; *Dary, 2011*). The Sandell-Kolthoff reaction was used to rate the reduction reaction of iodine from ceric ammonium sulfate (yellow) to the cerous form (colorless). Each standard's iodine concentration was graphed on the abscissa against its

optical density of 405 μg/l (OD405) on the ordinate. These results were all interpreted by making a standard curve on graph paper (*Machado et al., 2017*). The recommended procedure is as follows:

1. Each urine sample was mixed to suspend the sediment.
2. Each urine sample (250 mL) was pipetted into a 13 × 100 mm test tube. Each iodine standard was pipetted into a test tube, and then H2O was added as needed to make a final volume of 250 ml. Duplicate iodine standards and a set of internal urine standards were used in each assay.
3. Subsequently, 1 ml of 1.0 M ammonium sulfate was added to each tube.
4. All the tubes were heated for 60 min at 100 °C and then cooled to room temperature.
5. A 2.5-ml solution of arsenic acid was added. Each sample was mixed by flipping it up and down and left to right, then standing for 15 min.
6. Approximately 300 ml of ceric ammonium sulfate solution was added to each tube (quickly mixed) at 15–30 s intervals between successive tubes.
7. A stopwatch is used for this purpose. In practice, a 15-second interval is convenient.
8. The samples were allowed to stand at room temperature. Thirty minutes after the addition of ceric ammonium sulfate to the first tube, absorbance was read at 420 nm. Successive tubes were read at the same intervals as those when ceric ammonium sulfate was added. The results were determined using the World Health Organization reference for the cut-off value of iodine adequacy. Thus, the median UIC of 100 μg/L and higher values was substantially adequate, and an iodine concentration under 100 μg/L was classified as inadequate. In addition, a UIC of 50 to 99 μg/L is insufficient, and under 50 μg/L indicates severe iodine deficiency (*Dary, 2011*).

### *Anthropometry measurement*

ENA software was used to generate a height-for-age Z-score (HAZ), and an HFA >-2 z-score was indicated as a growth defect. Growth defects in children were classified as very high (>40%), high (30–39%), medium (20–29%), or low (<20%) (*Perumal, Bassani & Roth, 2018*).

### Data processing and analysis

Data were analyzed using the IBM Statistical Package for Social Science version 28 (SPSS; Chicago, IL, USA). An independent $t$-test was used to compare baseline mean differences in iodine concentrations between the intervention and control groups. Finally, generalized estimation equations (GEE) were used to predict all necessary variables useful for predicting independent variables (*Kent State University, 2020*). The results were further interpreted using beta coefficients ($\beta$) and 95% confidence intervals. Subsequently. For the $t$-test, a two-sided $p$-value was used to show statistical differences between the intervention and control groups as well as the significance of the GEE associations.

### Ethical approval

The Jimma University Institutional Review Ethics Board (IRth) granted approval for our research on January 1, 2018 (reference number IHRPGD/3007/18).

### Informed consent and confidentiality

Informed consent was obtained from the mother or caregiver of each child. Each mother or caregiver was informed of the need for voluntary participation in the study.

## RESULTS

### Follow-up success

In the first stage of this study, 16 LCUs selected from different districts were enrolled and randomized into eight intervention clusters and eight control groups. Each group included 417 mothers, caregivers, and paired children selected from each LCU in proportionate numbers. At baseline, 834 (100) children completed their follow-up and participated in the end-line survey without loss of follow-up (Fig. 2).

### Urine iodine excretion results

A total of 195 children (46.8%) in the control group and 208 children (49.9%) in the intervention group were younger than 24 months of age, and those aged 24 months or older provided urine samples for examination. By the end of the study, the average age of the children was 38.60 months (+/−12.8 SD) compared to the baseline of 27.4 months (+/− 13.1 SD). However, only 66 (15.8%) children from the control group and 84 (20.1%) from the intervention group were under 24 months of age, 684 (82%) were older than 24 months, and children from both groups provided urine samples at the end of the study. In fact, 574 (68.8%) children adhered to their growth monitoring, and 406 (48.7%) were female. Of the mothers, 159 (19.1%) were illiterate, and 626 (75%) were older than 35. Mothers knew a lot (72%) about adding iodized salt to food at the end of cooking, and 183 (21.9%) of them did so regularly and purposefully. The mean urine iodine concentration (UIC) was 107.7 (+/− 8.64 SD) and 260.9 (+/− 149 SD) at the baseline survey and the end-line survey, respectively. Inadequate iodine excretion in the urine of children was present at baseline in 127 (15.2%) cases, which was nearly equal in both the control and intervention groups ($n = 63$ and 64, and 15.1% and 15.2%, respectively). However, iodine deficiency was reduced to 63 (7.6%) at the end of the study and critically reduced in the intervention group ($n = 11$, 2.6%), but only marginally changed in the control group ($n = 52$, 12.5%). The median UIC showed the undeniable change that its median increased from 107.7 μg/L to 209.9 μg/L, with the percentiles of 25 and 75 being 143 and 400 μg/L, respectively. The children who participated in this study had a mean height (Ht) of 83.1 cm (+/−10 SD) at baseline and 136.4 cm (+/−14 SD) at the end of the survey. Stunted growth at baseline was common, with 169 children (40.5%) in the intervention group and 171 children (41.0%) in the control group. In conclusion, the intervention group ($n = 63$, 15.1%) significantly outperformed the control group ($n = 213$, 51.1%) (Table 1).

### Results of the iodized salt analysis

The iodide level was above or equal to the advised parts per million (ppm) in 164 (98.1%) of the table salt samples, with a mean of 45.3 ppm and a standard deviation (SD) of 14.87. The analyzed table salt contained very high levels of iodine. Of the 164 table salt samples, 98.2% had over or at the standard parts per million (ppm) iodine, and three (1.8%) samples

**Table 1** Socio-demography character of households and nutritional status of children.

| Variables | n = 834 | Control (N = 417) Frequency (%) | Intervention (N = 417) Frequency (%) |
|---|---|---|---|
| Mothers' age | <20 years | 8 (2.0) | 7 (1.7) |
| | 20 to 35 years | 100 (24.0) | 93 (22.3) |
| | <35 years | 317 (76.0) | 309 (74.0) |
| Household income/104 | <1040 Bir | 206 (49.4) | 156 (37.4) |
| | 1040 to 2080 Bir | 101 (24.2) | 93 (22.3) |
| | >2080 Bir | 110 (26.4) | 168 (40.3) |
| Children gender | Female | 208 (49.9) | 199 (47.7) |
| | Male | 209 (50.1) | 218 (52.3) |
| Baseline children age | <24 months | 195 (46.8) | 208 (49.9) |
| | >24months | 222 (53.2) | 209 (50.1) |
| End-line children | <24months | 66 (15.8) | 84 (20.1) |
| | >24months | 351 (84.2) | 333 (79.9) |
| UIC at baseline | <100 | 63 (15.1) | 64 (15.2) |
| | >100 µg/L | 354 (84.9) | 353 (84.7) |
| UIC at end-line | <100 | 52 (12.5) | 11 (2.6) |
| | >100 µg/L | 365 (87.5) | 406 (97.4) |
| Baseline HFA | Z-score <-2SD | 171 (41.0) | 169 (40.5) |
| | Z-Score >-2SD | 246 (59.0) | 248 (59.5) |
| End-line HFA | Z-Score <-2SD | 213 (51.1) | 63 (15.1) |
| | Z-Score >-2SD | 204(48.9) | 354 (84.9) |

**Notes.**
HFA, Height for age; µg/L, Microgram per litter of urine; N, sample in group; UIC, Urine iodine concentration.

had less than 15 ppm iodine. The average weight of the table salt samples used for analysis was 10 g. During table salt analysis for iodine content, the final and initial volumes in ml were symmetrical to the difference in volume/ml (Fig. 3).

## Differences in UIC among groups and other variables

There were no differences in iodine deficiency (UIC < 100 µg/L) between children assigned to the interventional 64 (15.3%) and control 63 (15.1%) at baseline. However, at the end of this study, the intervention group had the lowest proportion (11 (2.6%)) and the control group had 52 (12.5%), $p = 0.001$.

In terms of stunting growth (height for age (HFA) - 2SD), the children assigned to the intervention group and the control group had comparable higher rates at baseline ($n = 69$, 40.5%) and 171 (41%), respectively. At the end of the study, the higher percentage of stunting growth fell to lower levels ($n = 63$, 15.1%) among the intervention group and reached a higher level than the baseline ($n = 213$ (51.1%) among the control group (Table 2).

## Results from an independent *t*-test analysis

The mean difference (MD) in urine iodine concentration (UIC) between the intervention and control groups was 97.56 µg/L, with a standard error (SE) of 9.83 ($p = 0.001$). Additionally, with an MD of 21.78 µg/L UIC (SE 10.38, $p = 0.036$), female children

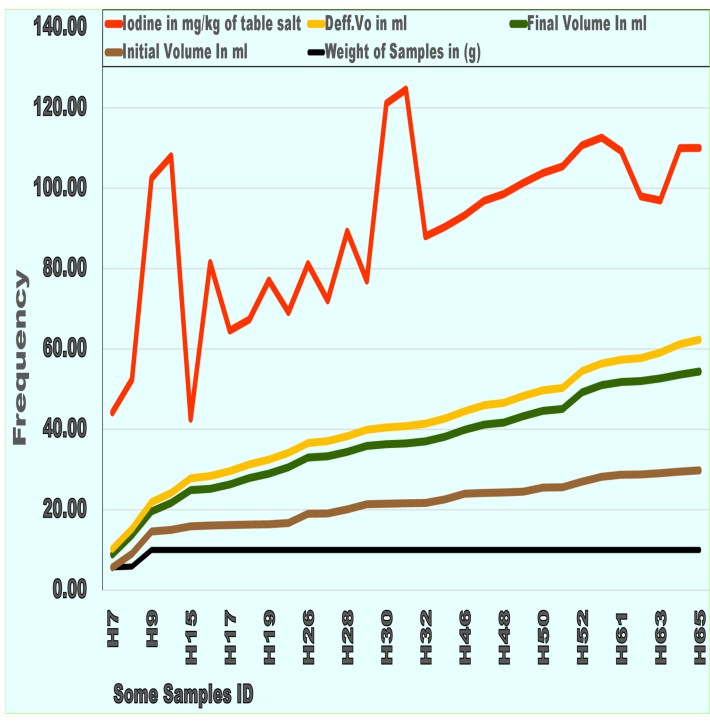

**Figure 3  Some of table salt samples laboratory analysis on quantify iodide results at lower community units and their markets.** Iodine mg, iodine in mg per kg of table salt; Def.Vo ml, Differential volume in ml in analysis; Final .Vo ml, Final volume in ml in analysis; Init .Vo ml, Initial volume in ml in analysis; Wt of salt, Weight of salt Sample in gram.

were better protected from iodine deficiency than male children. Similarly, the MD of iodized salt use during food preparation, which is typically performed with awareness, was 29.42 μg/L (SE 14.56, $p = 0.045$). Children's growth with and without stunting was found to differ significantly in the MD of UIC (MD = 49.06 g/L, SE = 10.69, $p = 0.001$), as were households with monthly incomes of 208 Birr or more and those with lower incomes (MD = 26.41 g/L, SE = 11.32, $p = 0.02$).

Children whose mothers continuously added iodized salt during later cooking times had a higher MD of UIC (91.38 μg/L) (SE = 11.19, $p = 0.001$). Children who had a height forage (HFA) Z-score equal to or greater than -2 SD had a 49.06 μg/L UIC mean difference (SE = 10.09, $p = 0.001$) in comparison to those whose HFA z-score was less than -2 SD (Table 3).

## Results from the GEE analysis

The final result from the GEE analysis was that being part of the intervention group resulted in a height increase of 7.93 cm (3 = 7.93, $p = 0.005$), which was a positive aspect of increasing the end-line height (Ht) of the children. In addition to this factor, baseline height only by 0.36 cm ($\beta = 0.36$, $p = 0.001$) and baseline iodine status by 3.3 cm ($\beta = 1.57$, $p = 0.001$) were more likely to increase the children's height than all other parameters. For the mother models, there was a significant difference between mothers with a higher

**Table 2  Variables competition amid intervention and control groups at baseline and end-line survey.**

| Variables | Baseline, $n = 834$ Intervention Control $n = 417$ [%] $n = 417$ [%] $P$ | End line, $n = 834$ Intervention Control $n = 417$ [%] $n = 417$ [%] $P$ |
|---|---|---|
| **Age** | | |
| >24 Months | 209 (50.1) 222 (53.2) | 333 (79.9) 351(84.2) |
| <24 Months | 208 (49.9) 195 (46.8) 0.883 | 84 (20.1) 66 (15.8) 0.105 |
| **UIC** | | |
| >100 µg/L | 353 (84.7) 354 (84.9) | 406 (97.4) 365 (87.5) |
| <100 µg/L | 64 (15.3) 63 (15.1) 0.98 | 11 (2.6) 52 (12.5) 0.001 |
| **HAZ** | | |
| >-2SD | 248 (59.5) 246 (59.0) | 354 (84.9) 204 (48.9) |
| <-2SD | 169 (40.5) 171 (41.0) 0.888 | 63 (15.1) 213 (51.1) 0.001 |
| **WAZ** | | |
| >-2SD | 361 (86.6) 363 (87.0) | 311 (74.6) 267 (64.0) |
| <-2SD | 56 (13.4) 54 (13.0) 0.838 | 106 (25.4) 150 (36.0) 0.766 |
| **Use iodized salt** | | |
| With awareness | 64 (15.3) 81 (19.4) | 255 (61.1) 158 (37.9) |
| Without | 353 (84.5) 336 (80.6) 0.643 | 162 (38.9) 259 (62.1) 0.452 |
| **Storing table salt** | | |
| Formal dry place | 246 (78.26) 330 (79.1) | 365 (87.5) 349 (83.7) |
| Informal place | 171 (21.74) 87 (20.9) 0.211 | 52 (12.5) 68 (16.3) 0.115 |
| **AISUU in HH** | | |
| Yes | 293 (70.3) 314 (75.3) | 344 (82.5) 344 (82.5) |
| No | 124 (29.7) 103 (24.7) 0.861 | 73 (17.5) 73 (17.5) 0.001 |

**Notes.**
HAZ, Height for Age Z-Score; Median UIC, Median Urine Iodine Concentration; $n$, Number of study subjects; WAZ, Weight for Age Z-Score; UUAIS In HH, Adequately iodized salt usually used in the household; $P$, significance of differences.

educational status associated with their children who grew taller by 1.35 cm ($\beta = 1.35$, p 0.001) and mothers with a lower educational status. The parameters of animal food intake influenced children's height by 6.83 cm ($\beta = 6.83$, $p = 0.001$), which was associated with their dietary intake behavior. Additionally, consumption of fruit by children after meals increased by 5.39 cm ($\beta = 5.39$, $p = 0.001$) and regular use of iodized salt increased by 0.88 cm ($\beta = 0.88$, $p = 0.001$), both of which significantly increased the height of children. According to the knowledge model for iodine, using suitable salt containers in homes to preserve iodized salt from moisture is more likely to result in children growing taller by 2.53 cm ($\beta = 2.53$, $p = 0.001$). Table 4 presents the full GEE results.

## DISCUSSIONS

The majority (98.2%) of table salt samples from homes and markets had iodine concentrations above the recommended levels, but children from these homes had a moderate (15.2%) iodine deficiency. More than 30% of the iodine loss from salt production to consumption can be attributed to the gap between iodized, adequately contained table salt and children's iodine concentrations (*WHO, 2014*). Additionally,
**Table 3  The difference of end-line—baseline mean UIC among the groups and other variables.**

| Variables | $n = 834$ (%) | Mean of UIC | UIC MD | Std. Error Difference | $p$-values |
|---|---|---|---|---|---|
| **Groups** | | | | | |
| Intervention | 417 (50) | 202.0 | | | |
| Control | 417 (50) | 104.4 | 97.56 | 9.83 | 0.001 |
| **Children gender** | | | | | |
| Male | 428 (51.3) | 142.6 | | | |
| Female | 406 (48.7) | 164.4 | −21.78 | 10.38 | 0.036 |
| **End-line children age** | | | | | |
| <24 months | 684 (82.0) | 151.0 | | | |
| >24 months | 150 (18.0) | 163.23 | −12.2 | 13.21 | 0.359 |
| **Mothers' education** | | | | | |
| Low education level | 698 (83.7) | 153.19 | | | |
| High school | 136 (16.3) | 153.35 | −0.15 | 14.33 | 0.99 |
| **Household income** | | | | | |
| 208 Bir or more/month | 259 (31.1) | 171.43 | | | |
| Lesser 208 Bir/month | 575 (68.9) | 145.01 | 26.41 | 11.20 | 0.019 |
| **Use iodized salt** | | | | | |
| With awareness | 145 (17.4) | 177.53 | | | |
| Without awareness | 689 (82.6) | 148.10 | 29.42 | 13.68 | 0.032 |
| **Adding iodized salt** | | | | | |
| Early cooking time | 230 (25.6) | 219.40 | | | |
| Later cooking time | 604 (74.4) | 128.02 | 91.38 | 11.19 | 0.001 |
| **Micronutrient rich foods** | | | | | |
| Eat usually | 693 () | 151.85 | | | |
| I don't know it | 141 () | 159.94 | −8.09 | 13.87 | 0.056 |
| **Baseline HFA Z-Score** | | | | | |
| <-2SD | 340 (40.8) | 156.73 | | | |
| >-2SD | 494 (59.2) | 150.73 | −6.10 | 10.44 | 0.560 |
| **End-line HFA Z-Score** | | | | | |
| >-2SD | 556 (66.9) | 169.45 | | | |
| <-2SD | 276 (33.1) | 120.39 | 49.06 | 10.09 | 0.001 |

**Notes.**
HFA, Height for age; MD, Mean difference; $n$, sample size; SD, Standard deviation; UIC, urine iodine concentration.

when the temperature was humid, when it was opened in the light, or when it was not kept in the dark, the iodine content of the table salt could be reduced. However, the best conditions for preserving iodized salt are room temperature, complete darkness, and a lack of humidity (*Fallah et al., 2020*). This behavior change intervention doubled the mean iodine concentration within the intervention group by introducing the communication behavior change needed for handling and using iodized salt at the household level. It also aimed to close the knowledge gap about iodized salt utilization and enhance utilization techniques. More changes were observed in the very high prevalence of stunted growth in children, which decreased to a medium prevalence (15.2%), in addition to the change in UIC. Additionally, the median UIC showed the undeniable change that the median UIC

**Table 4 The results of the GEE analysis for predicting children's heights at the end of intervention are displayed as follow.** The asterisks (*) indicate that the results are statistically significant.

| Parameter | β | Std. Error | 95% Wald CI Lower | 95%Wald CI Upper | p-values |
|---|---|---|---|---|---|
| (Intercept) | 130.20 | 1.57 | 127.12 | 133.3 | 0.001* |
| Intervention group | 7.93 | 2.82 | 2.40 | 13.45 | 0.001 |
| Children gender | −2.22 | 2.47 | −7.07 | 2.64 | 0.001* |
| Household income | 1.30 | 0.29 | 0.73 | 1.86 | 0.715 |
| Food preparation | −0.98 | 2.69 | −6.26 | 4.29 | 0.262 |
| Iodine adequately used in HH | −2.98 | 2.65 | −8.18 | 2.22 | 0.016 |
| Introduced iodine communication | 0.80 | 0.34 | 0.15 | 1.46 | 0.314 |
| Starting time of complementary | 0.12 | 0.12 | −0.12 | 0.36 | 0.001 |
| Baseline iodine status (UIC) | 3.30 | 0.29 | 2.72 | 3.87 | 0.283 |
| Duration breastfed | −0.03 | 0.03 | −0.08 | 0.02 | 0.114 |
| Growth monitoring follow-up | −1.02 | 0.63 | −2.25 | 0.24 | 0.242 |
| Weight of children baseline | −.234 | 0.20 | −0.63 | 0.16 | 0.156 |
| Adequately iodized in food | 2.97 | 2.09 | −1.13 | 7.06 | 0.001* |
| Use salt container protect moisture | 2.53 | 0.07 | 1.74 | 3.32 | 0.001* |
| Animal food for children | 6.83 | 1.95 | 3.01 | 10.65 | 0.001* |
| Eat fruits after mail | 5.39 | 0.34 | 3.01 | 10.65 | 0.001* |
| Micronutrient intake necessary used | −1.89 | 0.34 | −2.56 | −1.22 | 0.001* |
| Adequately iodized salt used in HH | .877 | 1.14 | −1.35 | 3.10 | 0.440 |
| Education status of mother | 1.35 | 0.34 | 0.69 | 2.01 | 0.001* |
| Weight of children at baseline | −.510 | 0.32 | −1.13 | .109 | 0.106 |
| Height of children at baseline | .361 | 0.02 | 0.32 | .404 | 0.001* |
| HFA z-score at the baseline | .861 | 0.78 | −0.67 | 2.39 | 0.27 |
| Complementary food started time | .547 | 0.39 | −0.23 | 1.32 | .167 |
| (Scale) | 602.2 | | | | |

**Notes.**

β, Beta Estimate; UIC, urine iodine concentration; HFA, Height for age; HH, Households; Std. Error, Standard error; Wald CI, Confidence interval of odds ratio.

increased from 107.7 µg/L to 209.9 µg/L, with the percentiles of 25 and 75 being 143 and 400 µg/L, respectively. Therefore, early behavioral change interventions can stop children's slow growth and development. Additionally, enhancing mothers' knowledge and attitudes facilitated children's dietary intake, accelerated the transition of their iodine status from incomplete to secure, and decreased the percentage of children who failed to thrive in the child population (*Kulik et al., 2019*). This kind of intervention is crucial for this community because the majority of people in Ethiopia are uneducated and underprivileged (*UNICEF, 2019*). Furthermore, it is commonly known that wealth, educational attainment, and urban or rural status all have an impact on the likelihood of households using more iodized salt (*Karmakar et al., 2019*). As a result, many developments in nutrition and health sciences have not yet been sufficiently applied to behavior changes for the benefit of the community. Additionally, studies on behavior change have not provided a comprehensive body of advice for people and organizations wishing to align routine behavior with goals that will endure forever (*Duckworth & Gross, 2020*). Communities with socioeconomic and demographic

problems must give nutrition, behavioral change, and communication top priority to avert iodine deficiency and stunted physical growth (*Vaivada et al., 2020*). Additionally, behavioral modification is the most effective intervention strategy for boosting children's consumption of iodine, other nutrient-based foods, and high-quality diets (*McInerny, 2017*). In addition to determining whether households had sufficient iodine in table salt, this study sought to lower the moderate level of iodine deficiency detected in the baseline survey in young children. Communication regarding changes in dietary intake behavior addresses a variety of community-wide health and nutrition challenges. This intervention helped the intervention group's children grow and address their related problems. To date, no study has addressed the various aspects of iodized salt intervention monitoring and evaluation systems, which have contributed to the failure of intervention programs for many years (*Choudhry & Nasrullah, 2021*; *Adu & Simpong, 2017*). These are unlikely to protect young children from iodine-deficiency-related brain impairment. Additionally, an intervention program has been established, and numerous surveys from the past few decades have recommended activities for evaluating iodine deficiency disorders. The inability to holistically understand and treat iodine deficiency in young children has limited efforts to address issues such as brain damage, low IQ, high school attrition rate, and poor physical growth (*Andersson, Karumbunathan & Zimmermann, 2012*). Therefore, it may be possible to prioritize and direct interventions towards the most vulnerable groups by identifying children with iodine deficiency due to the underuse of iodized salt, which is associated with a variety of socioeconomic and democratic challenges (*Knowles et al., 2017*).

### Statement of significance

We concurred to participate in this study because, at the outset, children assigned to the control and intervention groups had moderately low levels of iodine deficiency, while table salt had very high levels of iodine. To close this gap, a focused nutrition and behavior change intervention was provided to mothers. For the past fifteen months, sufficient information about good food practices, the importance of iodine, and its handling and application has been given to mothers for a number of months. In addition to managing and using iodine salt, it has also introduced options for other healthy eating habits that will also play a significant role in their children's future development. This sort of knowledge transfer intervention is essential for the sustainability of society's health. Therefore, this trial's implications revealed that the intervention group's iodine status and growth could essentially be improved while the control group continued to experience negative effects. The merit of this project was rigorously providing nutritional behavior change to mothers, enabling us to adopt. Additionally, many specimen samples were carefully examined in a government laboratory, which increased the reliability and accuracy of the dataset.

## LIMITATION

We measured the iodine quantity found in table salt and children's UIC twice, but we could not determine the long-term effect of UIC except for physical growth change. Longitudinal studies are important for understanding this gap.

## ACKNOWLEDGEMENTS

We would like to thank the Ethiopian Public Health Institute for analyzing the iodized table salt samples and urine specimens. Our special appreciation is extended to the Arsi Zone administrative offices and the Zone Health Department for their assistance and cooperation during data collection and intervention periods. We also thank the health extension workers of the 16 rural LCUs and study participants. Finally, we thank the eight district health officer heads for their involvement in this study.

### Funding

This study was supported by Jimma University under grant number T/AI/M/D/M/-DA/3022/2018. The funders had no role in study design, data collection and analysis, decision to publish, or preparation of the manuscript.

### Grant Disclosures

The following grant information was disclosed by the authors:
Jimma University: T/AI/M/D/M/DA/3022/2018.

### Competing Interests

The authors declare there are no competing interests

### Author Contributions

- Abebe Ferede conceived and designed the experiments, performed the experiments, analyzed the data, prepared figures and/or tables, authored or reviewed drafts of the article, project and finance administration, and approved the final draft.
- Muluemebet Abera Wordofa conceived and designed the experiments, performed the experiments, analyzed the data, prepared figures and/or tables, authored or reviewed drafts of the article, and approved the final draft.
- Tefera Belachew conceived and designed the experiments, performed the experiments, analyzed the data, prepared figures and/or tables, authored or reviewed drafts of the article, and approved the final draft.

### Ethics

The following information was supplied relating to ethical approvals (*i.e.*, approving body and any reference numbers):

Jimma University Institutional Review ethics board (IRB) (reference number IHRPGD/3007/18).

### Data Availability

Ferede, Abebe, 2021, "Replication Data for: Micronutrient concentration effects on growth defect of children after Nutrition Behavior intervention in Central Highland of Ethiopia: Cluster Randomized Trial.", https://doi.org/10.7910/DVN/USJMUT, Harvard Dataverse, V1, UNF:6:5QpoQfplKZBWJ8FyFUF2qQ== [fileUNF]

## Supplemental Information

Supplemental information for this article can be found online at http://dx.doi.org/10.7717/peerj.16849#supplemental-information.

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
