# Peer review of "Behavior change intervention to sustain iodide salt utilization in households in Ethiopia and study of the effect of iodine status on the growth of young children: community trial"

_PeerJ, doi:10.7717/peerj.16849_

## Round 0.1 · original submission · Major Revisions

Thank you authors for your patience. Reviewers have considered your work, and raised very important concerns, despite noting the novelty of your work.
Please carefully address in detail all their concerns. Essentially, elaborate further on the statistical aspects of the methodology.
Look forward to your revised manuscript.

**Language Note:** The review process has identified that the English language must be improved. PeerJ can provide language editing services - please contact us at copyediting@peerj.com for pricing (be sure to provide your manuscript number and title). Alternatively, you should make your own arrangements to improve the language quality and provide details in your response letter. – PeerJ Staff

·

Basic reporting

1. Write The Objective of Study at The End of Introduction.
2. Grammatical errors need to be corrected.

Experimental design

2. Describe the setting (state or country in form of geography and demography)
3. Add full form of LCU in methods section where it is first used.
4. “To evaluate the iodide status of table salt found in households (HHs), approximately 834 table salt samples were collected, and 120 samples were selected using systematic random sampling for laboratory analysis. In addition, 25 samples were randomly selected from the 50 table salt samples collected from markets. Each LCU was represented by more than nine table salt samples.” Are the 120 sample part of 834? Are 50 from market part of 834? The line is not clear where and why systematic random sampling is applied. What was sampling frame from where was it obtained. How was the first study unit decided? Reword it.
5. “The selection of a few k from a large number of LCUs provided sufficient protection of the behavior change information from contamination with the control.” What is k? the line is not understandable. Reword it according to how masking was ensured.
6. Was any behavior change model used or imparting behavior change communication? If yes mention it
7. Was any wash out period provided? If not why
8. Who was the study unit? Children with mother or household? If it was child-mother pair, did you enroll all children from a single household or only one?
9. Write a line about HEW. In your system what type of workers are they?
10. Was the intervention provided in group or by one to one mode? Please mention in implementation section.
11. Mention the outcome variables in which you saw the change.

Validity of the findings

12. Line 166-168 not clear whether it is number of urine samples with iodine or total?Reword it
13. Line 188: What is full form of MD?
14. Line 201-203: some grammatical errors in interpreting the line, please reword
15. The text inside fig 1 is not clear
16. Table 3: GEE analysis: Children gender was taken as categorical variable: what was reference? Check spellings too. Food preparation and cooking: what type of variable if categorical pls mention which one was ref. similarly present the table in away it is readable.

Additional comments

No comment

·

Basic reporting

Overall, it is hard to follow this manuscript because the writing is unclear and have many grammar errors. The manuscript needs extensive english editing.

Experimental design

This study is novel and original. The knowledge gap is well-defined. Rigorous investigation was performed. However, the statistical analyses were not described or performed sufficiently.

- In section 2.8, *Data processing and analysis, please provide more details on what descriptive statistics were used in Table 1. For example, “Means and standard deviations were used to describe continuous variables, and frequencies and proportions were used to describe categorical variables.” Please also compare the baseline variables between the interventional and the control groups to show whether the randomized controlled trial has achieved the desired outcome - no significant differences in participant characteristics between intervention and control at baseline. Also, please indicate what criteria were used to determine a statistical significance. For example, “a two-sided p-value was used to indicate statistical significance.”

- In lines 148-149, you mentioned that, “An independent t-test was used to compare baseline mean differences in iodine concentrations between the intervention and control groups.” But based on table 1, I think you actually compared the mean difference in UIC from baseline to the endpoint between the intervention and control groups and the mean differences between other covariates’ categories. So please rephrase accordingly.

- In lines 149-150, you mentioned that the variables with a p-value < 0.25 were included in the GEE analysis. This part is very confusing. I think in table 3, the outcome variable is height, in table 2, you tested the associations of mean UIC change with other covariates instead of the association of height with other covariates. In addition, some of the covariates included in Table 3 GEE analysis are not shown in Table 2. Could you elaborate on your variable selection process for the GEE analysis predicting children's height? In addition, have you done any model diagnosis or goodness-of-fit test for the final GEE model?

Validity of the findings

no comment

Additional comments

no comment

---

## Round 0.2 · Minor Revisions

Please, authors, the reviewers have considered your work favorably. However, some work is still required to improve the quality of work.
Kindly address the concerns raised, and where applicable, provide direction/recommendations for future studies in your conclusions.

·

Basic reporting

NO COMMENTS

Experimental design

NO COMMENTS

Validity of the findings

NO COMMENTS

Additional comments

The majority of comments have been dealt with.

·

Basic reporting

no comment

Experimental design

- In lines 233-234, please also indicate how you selected those covariates that were included in the GEE model
- In table 4, either p-value or the confidence interval was mismatched to the covariates. For example, the confidence interval for the "Intervention group" is (2.40 13.45) but the p-value is over 0.05.
- Several variables in the GEE model have very non-significant p-values. Why did you include them in the model in the first place? What do your model's goodness of fit metrics look like? Could you conduct a model diagnosis to show if this GEE model fits your data?

Validity of the findings

no comment

---

## Round 0.3 · accepted · Accept

Thank you authors for revising your work, and addressing the concerns raised. The reviewers' comments helped to elevate the quality of the work. It is now acceptable for publication. Thank you for finding PeerJ as your journal of choice, and look forward to your future scholarly contributions. Congratulations :)

·

Basic reporting

no comment

Experimental design

no comment

Validity of the findings

no comment

Additional comments

no comment